# Cross-Examination of Similarity, Difference and Deficiency of Gauge, Radar and Satellite Precipitation Measuring Uncertainties for Extreme Events Using Conventional Metrics and Multiplicative Triple Collocation

**Zhi Li** [1], **Mengye Chen** [1], **Shang Gao** [1], **Zhen Hong** [1], **Guoqiang Tang** [2,3], **Yixin Wen** [4,5], **Jonathan J. Gourley** [4] **and Yang Hong** [1,*]

[1] School of Civil Engineering and Environmental Science, University of Oklahoma, Norman, OK 73072, USA; li1995@ou.edu (Z.L.); mchen15@ou.edu (M.C.); shang.gao@ou.edu (S.G.); Zhen.Hong-1@ou.edu (Z.H.)

[2] Coldwater Laboratory, University of Saskatchewan, Canmore, AL T1W 3G1, Canada; guoqiang.tang@usask.ca

[3] Center for Hydrology, University of Saskatchewan, Saskatoon, SK S7N 1K2, Canada

[4] NOAA National Severe Storms Laboratory, Norman, OK 73072, USA; berry.wen@noaa.gov (Y.W.); jj.gourley@noaa.gov (J.J.G.)

[5] Cooperative Institute for Mesoscale Meteorological Studies, Norman, OK 73072, USA;

[*] Correspondence: yanghong@ou.edu

**Abstract:** Quantifying uncertainties of precipitation estimation, especially in extreme events, could benefit early warning of water-related hazards like flash floods and landslides. Rain gauges, weather radars, and satellites are three mainstream data sources used in measuring precipitation but have their own inherent advantages and deficiencies. With a focus on extremes, the overarching goal of this study is to cross-examine the similarities and differences of three state-of-the-art independent products (Muti-Radar Muti-Sensor Quantitative Precipitation Estimates, MRMS; National Center for Environmental Prediction gridded gauge-only hourly precipitation product, NCEP; Integrated Multi-satellitE Retrievals for GPM, IMERG), with both traditional metrics and the Multiplicative Triple Collection (MTC) method during Hurricane Harvey and multiple Tropical Cyclones. The results reveal that: (a) the consistency of cross-examination results against traditional metrics approves the applicability of MTC in extreme events; (b) the consistency of cross-events of MTC evaluation results also suggests its robustness across individual storms; (c) all products demonstrate their capacity of capturing the spatial and temporal variability of the storm structures while also magnifying respective inherent deficiencies; (d) NCEP and IMERG likely underestimate while MRMS overestimates the storm total accumulation, especially for the 500-year return Hurricane Harvey; (e) both NCEP and IMERG underestimate extreme rainrates (>= 90 mm/h) likely due to device insensitivity or saturation while MRMS maintains robust across the rainrate range; (g) all three show inherent deficiencies in capturing the storm core of Harvey possibly due to device malfunctions with the NCEP gauges, relative low spatiotemporal resolution of IMERG, and the unusual "hot" MRMS radar signals. Given the unknown ground reference assumption of MTC, this study suggests that MRMS has the best overall performance. The similarities, differences, advantages, and deficiencies revealed in this study could guide the users for emergency response and motivate the community not only to improve the respective sensor/algorithm but also innovate multidata merging methods for one best possible product, specifically suitable for extreme storm events.

**Keywords:** extreme events; IMERG; MRMS; NCEP; triple collocation

## 1. Introduction

Extreme rainfall events associated with flash floods [1,2], landslides [3,4], and debris flow [5] often lead to tremendous damages to properties and fatalities. According to Cerveny [6] and Mazzoglio [7], extreme rainfall events are becoming more frequent. It is thus an increasing concern for emergency agencies, researchers, and the public, especially residents in coastal cities [8]. Tropical cyclones carry a considerable amount of water vapor from oceans inland and are the predominant cause of deaths in the United States and disruption to transportation, utilities, communications, and agriculture [8–10]. Hurricane Harvey, starting as a tropical storm, became a category four hurricane while making landfall in Texas causing devastating urban flooding and fatalities in August 2017. Based on the survey by Emanuel [11], Hurricane Harvey had reached the heaviest rainfall record in the history of the United States and caused at least 70 casualties, and an economic loss beyond 150 billion US dollars [11,12]. In Fall 2019, another tropical cyclone Imelda made landfall in Texas, with a recorded 1096 mm of rainfall that occurred in Jefferson County, which is ranked as the fourth-highest rainfall record in the history. Two additional tropical cyclones Bill and Cindy, influenced southeast Texas and brought in copious amounts of rainfall in 2015 and 2017, respectively.

Traditionally, rainfall is estimated by gauges as a direct measurement but only at a point scale [13]. Even though gauge data are often treated as the "ground truth" for rainfall measurement, they are still not impeccable due to splash-out during heavy rainfall, lack of sensitivity to light rain rates, under-catching by wind drift, and evaporation [14–16]. Especially in heavy rain events, multiple studies have demonstrated that the error caused by these inherent factors is not trivial [15,16]. Luyckx [15] investigated the disadvantages of tipping bucket rain gauges during extreme weather conditions and found there was an underestimation of rainfall volumes due to loss of water during the tipping action from the comparison of 24 well calibrated gauges. Molini et al. [16] quantified the bias as an underestimation from 60% to 100% for the 1h design rainfall and return periods from 20 to 200 years. The wind from tropical cyclones would substantially affect the performance of rain gauges as well, with the relative bias ranging from 5% to 80% [17]. Besides the systematic error, when interpolating point samples, errors caused by spatial interpolation accounts for 50% to 80% of the total difference depending upon the gauge quality and density [14,18]. Stampoulis and Anagnostou [19] discovered that the convective nature of rainfall can also increase gauge-interpolation uncertainties. All these may result in problems when one applies gauge-based corrections for other products, for example, radar QPE, satellite precipitation estimation in extreme events.

In the past three decades, the emerging radar technology has been applied in meteorology to estimate rainfall by emitting and receiving electromagnetic signals. The most prominent advantage of radar over rain gauges is that radar provides a more refined spatiotemporal scale and a larger area coverage. However, since it is an indirect measurement of rain rate, radar technology itself produces and propagates errors in its end products. The uncertainties of radar-based precipitation estimation can be categorized as incorrect calibration, sampling representativeness, non-weather echoes, and uncertainties in Z-R relations [14,17,20–23]. These inherent systematic errors are challenging to mitigate by improving radar technology alone. Thus, many researchers chose to blend radar with rain gauges and satellite data to improve the performance, for example, Kriging with External Drift (KED) [24,25], Mean Field Bias Correction (MFB) [26]. A few studies investigated the bias of radar rainfall products in excessive rainfall events [17,20,27,28]. Kidd et al. [28] compared ground-radar Quantitative Precipitation Estimation (QPE) with gauge data in Germany, and the results indicated radar overestimation in convective rainfall regimes. Medlin et al. [17] evaluated National Weather Service (NWS) Weather Surveillance Radar-1988 Doppler (WSR-88D) during Hurricane Danny. They concluded that both radar and rain gauge seriously underestimated rainfall. Cao et al. [20] evaluated the performance of S-band dual-polarized radar in Hurricane Irma in which the radar showed nearly 50% of underestimation. Gao et al. [29] evaluated the performance of Multi-Radar Multi-Sensor (MRMS) QPE during Hurricane Harvey. MRMS QPE underestimated the total accumulated rainfall by a small factor and overestimated very light precipitation.

Another commonly used data source is remote sensing satellite, attributing to the advantages of broad spatial coverage and an ability to scan downward over complex terrain [13,30]. Satellite-based precipitation products utilize the information provided by visible-infrared (VIS-IR) channels from geostationary (GEO) satellites, passive microwave (PMW) sensors, and spaceborne radars from low orbiting (LEO) satellites. Just like weather radar QPE, satellite rainfall retrieval is also an indirect measurement of rainfall. Many previous works [12,30,31] stated that satellite data would underestimate the magnitude of high rainfall rates. Hong et al. [32] stated that satellite performance degrades with the increase of rain rate but has higher relative bias in the light rain. Chen et al. [27] compared four satellite precipitation products (SPPs) and ground-based radar with gauge references for Typhoon Morakot, and they found that SPPs underestimate extreme rainfall. Omranian et al. [12] evaluated Integrated Multi-satellitE Retrievals for GPM (IMERG) Version 5 final gauge-adjusted products with the National Weather Service/National Centers for Environmental Prediction (NWS/NCEP) River Forecast Center (RFC) Stage-IV Quantitative Precipitation Estimates (QPEs) as the reference in Hurricane Harvey. They revealed that IMERG is able to detect the spatial variability of the rainfall field but suffered in heavy rainrate regions.

Given the context that these independent data sources come with its respective inherent deficiencies, researchers performed stochastic approaches to analyze these uncertainties [13,33]. Triple Collocation (TC) has been proven to be a powerful statistical tool to estimate uncertainties within each of three independent products [34–42]. TC was firstly applied to evaluate ocean surface wind variability by inputting different wind products [41]. After that, it has been extended to determine errors of sea surface temperature [36], sea surface salinity [43], and wave height [35]. Roebeling et al. [40] were the first to apply TC in hydrometeorology to accommodate remote sensing, weather radar and rain gauges in Europe. Massari et al. [38] compared the performance of five satellite precipitation products over the U.S., and applied the TC method to evaluate correlation coefficients worldwide. Alemohammad et al. [34] introduced the Multiplicative Triple Collocation method (MTC), suggesting its appropriateness in rainfall error estimation and proposed a way to decompose the error term in order to investigate the violation of assumptions. Li et al. [37] used the TC method to perform uncertainty analysis for five SPPs, reanalysis data, and gridded gauge data over ungauged regions in Tibetan Plateau in China after validating TC with traditional statistics. Tang et al. [44] first evaluated the accuracy of snow observations for satellite precipitation products and reanalysis products.

To our knowledge, the TC and particular MTC methods have not been applied for evaluating gauge-radar-satellite precipitation products, exclusively for extreme events. Thus, the overarching goals of this study are to: (a) evaluate the applicability of MTC method in uncertainty assessment under extreme events; and (b) investigate the similarities, differences, and deficiencies of each product along with its inherent deficiency in extreme events. More specific objectives of this study include:

1.  Evaluate the applicability of MTC in extreme events with the cross-examination of three products using traditional metrics;
2.  Further examine the stability of MTC method's performance in multiple extreme events;
3.  Understand gauge rainfall product uncertainties in extreme events, which are often associated with splash-out, wind undercatch, as well as interpolation uncertainties;
4.  Understand satellite QPE uncertainties in extreme events, which are associated with signals that are indirectly tied to surface precipitation and poor spatiotemporal resolutions;
5.  Understand radar QPE uncertainties in extreme events, which are associated with incorrect Z-R formulations, non-weather signals, inadequate sampling;

This article is organized into four sections: Section 2 introduces the study domain and briefly reviews the three state-of-the-art datasets being used in this study; Section 3 describes in detail the formulas to derive Root Mean Squared Error (RMSE) and Correlation Coefficient (CC) from the TC method; Section 4 analyzes the results from a broad overview and further examines the specific event (Hurricane Harvey); and Section 5 provides the conclusions and future works.

## 2. Materials and Methods

### 2.1. Study Domain

The area of interest is in the Gulf Coast of the U.S., where it has endured several events recently and historically, as shown in Figure 1. It is one of the most frequently impacted areas by hurricanes, such as Hurricane Dennis (2005), Isaac (2012), Harvey (2017), and tropical storms such as Bill (2015), Cindy (2017), Imelda (2019) also affected this area in the past five years. Given that this study focuses on the GPM era, Figure 1 illustrates the storm tracks and accumulative precipitation amount of Bill, Cindy, and Imelda as well as Harvey, all after the year of 2015. Due to the unprecedented nature of Hurricane Harvey, the three other tropical storms were selected for analysis to improve the generality of the study findings. All three tropical storms were accumulated in total to match the Hurricane Harvey, which gave the analogous data sample sizes and accumulative rainfall amounts for two comparable study groups: Harvey and Other. The impacted area of the aforementioned events contains the states of Texas, Oklahoma, Louisiana, Arkansas, Tennessee, Mississippi, and Alabama, which almost accounts for 10 percent of the conterminous United States (CONUS). These events had similar moving patterns in that they approached inland from the Gulf of Mexico and then bending towards the northwest after making landfall, except for Imelda, which dissipated shortly after making the landfall. Table 1 lists the details of four individual events, including their durations and maximum amount of rainfall. Harvey was observed to contiguously produce rainfall across seven days. The durations of the other three tropical cyclones (Bill, Cindy, Imelda) were as short as three days, two days, and four days, respectively.

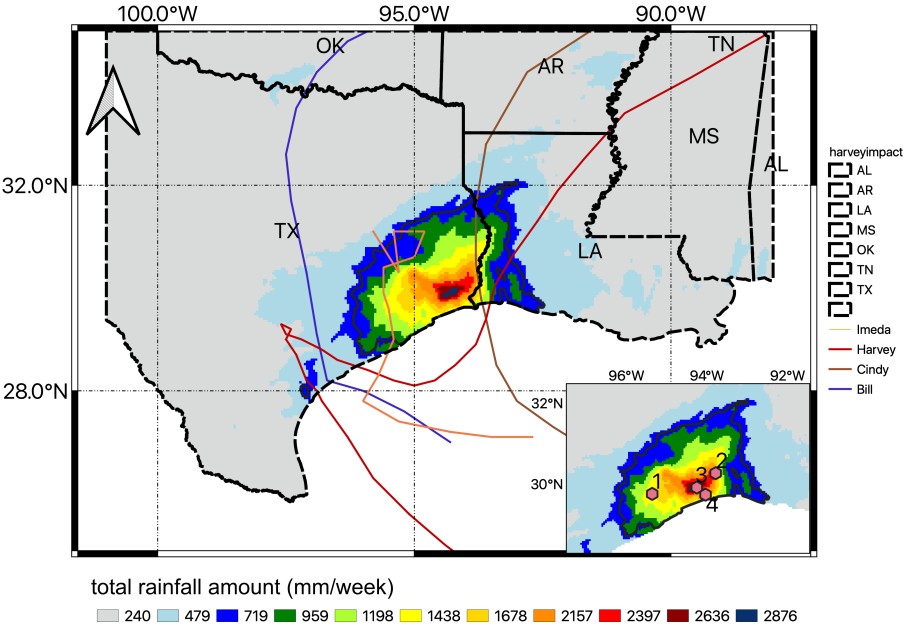

**Figure 1.** Accumulative rainfall for four events combined is measured by Multi-Radar Multi-Sensor (MRMS) radar data. Four cyclone tracks are illustrated with lines. Red dots (numbers from 1 to 4) in the bottom right panel are selected representative pixels in the Harvey core region.

**Table 1.** Events overview.

| Hurricane/Storm | Start Date | End Date | Duration | Maximum Rainfall Amount |
|---|---|---|---|---|
| Harvey | 25 August 2017 | 31 August 2017 | 7 days | 1625 mm |
| Bill | 16 June 2015 | 18 June 2015 | 3 days | 496 mm |
| Cindy | 22 June 2017 | 23 June 2017 | 2 days | 233 mm |
| Imelda | 18 September 2019 | 21 September 2019 | 4 days | 1126 mm |

## 2.2. Datasets Description

The NCEP gridded gauge-only hourly precipitation product (hereafter denoted as NCEP) is an operational product [45] covering the CONUS and parts of Puerto Rico. It is automatically derived from approximately 3000 operational hourly rain gauge observations across 48 states to produce a 4 km/hour rainfall field, using the interpolation method from Seo [46] in which the Double Optimal estimation (DO) and Single Optimal estimation (SO) are used to gain a conditional expectation of the rainfall estimation. This technique accounts for fractional coverage of rainfall due to sparse gauge networks. Gourley et al. [45] performed inter-comparisons of NCEP, NCEP Stage IV radar QPE, and PERSIANN-CSS SPP, and demonstrated that NCEP delivers the best performance at longer time scale, for example, seasonal, daily. However, it encounters underperformance at a shorter time scale, i.e., one hour, especially for the convective scale storms. NCEP data were obtained from the National Center for Atmospheric Research/Earth Observing Laboratory (NCAR/EOL): https://data.eol.ucar.edu/dataset/21.088.

The MRMS radar-only product (hereafter MRMS) has around 180 integrated operational radars, including 146 S-band and 30 C-band radars, creating seamless 3D radar mosaic across the CONUS and Southern Canada at 1 km/2 min resolution [47]. It is selected because of the strict quality control, involving filtering out non-hydrometeor signals, corrections for anomalous propagation, beam blockage, vertical profile reflectivity (VPR) correction, and adaptive Z-R relations [47]. Despite the adoption of these rigorous quality control steps, it still suffers from uncertainties of common issues mentioned before. For our study, 1 km/2 min radar-based QPE were retrieved and aggregated by averaging to 4 km/hourly, in order to be compatible with NCEP. Historical MRMS radar-only QPE was downloaded at http://mtarchive.geol.iastate.edu/.

IMERG satellite precipitation final product V06 (hereafter IMERG [48]) is an integrated SPP from its core satellite GPM Core Observatory (GPM CO), microwave constellations, Infrared, and additional constellations, aiming at providing global coverage of rainfall field (90N-S from V06 onward) beyond its predecessor Tropical Rainfall Measurement Mission (TRMM). GPM CO has additional channels of dual-frequency precipitation radar (DPR) and GPM Microwave Imager (GMI), which are capable of detecting very light precipitation and falling snow [49]. It produces three stages: early run, late run, and final run with 4 h latency, 12 h latency, and 3.5 months latency, respectively, to accommodate different purposes. The early run provides near real-time brief observations with inter-calibrated satellite products primarily for operational forecasting, and the late run adds up the late coming high-quality PMW data and climatological calibration to serve for agricultural purposes. The final run compares the late run product with the Global Precipitation Climatology Project (GPCP), and adjust the factor to compensate for under/over-estimation [48]. To account for the independence, we selected the final run without calibration in this study. In order to perform pixel-wise analysis, we further downscaled IMERG data using a nearest neighbor approach and then accumulated it up to hourly. The current IMERG final run product V06 can be accessed at https://disc.gsfc.nasa.gov/datasets/GPM_3IMERGHH_06/summary?keywords=IMERG.

## 2.3. TC Evaluations

### 2.3.1. Assumptions

A few assumptions are needed to comply for the TC method: (1) the three datasets should be independent, for instance, they are derived from different instruments; (2) the errors of the datasets should be independent or unrelated, which is referred to as zero cross-correlation. (3) The expectation of error is treated as zero known as the unbiasedness, which is often described in a geostatistical analysis, for example, kriging. Yilmaz and Crow [50] conducted experiments on the TC errors due to the relevance of three products, and the results revealed that the more independent they are, the less TC induced error. It is essential to consider the relevance of the inputs in order to make the TC method more reliable [37]. The three precipitation products selected, i.e., gauge-only, radar-only, and satellites-only meet the above criteria well because they are derived from different and independent instruments.

2.3.2. Expressions

The basics of the TC method is to treat three independent datasets as equally important, and thus no bias is produced in between them. Since no ground truth values are assumed, the TC method then uses a linear combination of three products and affine transformed error model to derive RMSE and CC [42].

$$R_i = a_i + b_i G + \epsilon_i \tag{1}$$

where $R_i$ indicates each of the independent source data, $G$ is the "relative truth", $a_i$, $b_i$ correspond to the weights and biases to adjust, and $\epsilon$ represents the error for each product. Tian et al. [51] then proposed a way to transform the additive error model to multiplicative by logarithmic transformation, and it is proved to be more appropriate in rainfall error estimation [34,37,51]. Hence, the error model can be reformed as:

$$r_i = a_i + b_i g + \varepsilon_i \tag{2}$$

From that, the rain rate and error model can be derived by transforming back into linear combination so that it fits into the TC method.

$$r_i = a_i + b_i g + \epsilon_i \tag{3}$$

where $r_i$ is the logarithmic form of rain rate $R_i$, $a_i = \ln \alpha_i$ demonstrates the multiplicative error, $\epsilon_i = \ln \varepsilon_i$ indicates the residual error, and $b_i = \beta_i$ as the deformation error. From linear Equation (3), we are able to derive RMSE in the following set of equations based on the covariance of triples [39]:

$$\begin{cases} \sigma_{r1}^2 = Cov(r_1, r_1) - \frac{Cov(r_1,r_2)Cov(r_1,r_3)}{Cov(r_2,r_3)} \\ \sigma_{r2}^2 = Cov(r_2, r_2) - \frac{Cov(r_1,r_2)Cov(r_2,r_3)}{Cov(r_1,r_3)} \\ \sigma_{r3}^2 = Cov(r_3, r_3) - \frac{Cov(r_1,r_3)Cov(r_2,r_3)}{Cov(r_1,r_2)} \end{cases} \tag{4}$$

Because the model is transformed to be in additive form, these parameters, along with error, should also be in logarithmic form. In the analysis of Alemohammad et al. [34], they transformed error to linear scale by Taylor series expansion as first order approximation.

$$\sigma_{r_i}{}^2 = \left( \frac{\sigma_{R_i}}{\mu_{R_i}} \right)^2 \tag{5}$$

$$\sigma_{R_i} = \mu_{R_i} \sigma_{r_i} \tag{6}$$

$\sigma_{R_i}$, $\sigma_{r_i}$, $\mu_{R_i}$ here represent the RMSE in linear form, logarithmic form, and the mean of field. In doing so, the error field could be identified in linear scale, meaning the same unit of mm/hour as rain rate. McColl et al. [39] introduced a way to evaluate correlation coefficient (CC) from manipulating covariance matrices in which CC is expressed below as a set of equations.

$$\begin{cases} CC_1^2 = \frac{Cov(r_1,r_2)Cov(r_1,r_3)}{Cov(r_1,r_1)Cov(r_2,r_3)} \\ CC_2^2 = \frac{Cov(r_1,r_2)Cov(r_2,r_3)}{Cov(r_2,r_2)Cov(r_1,r_3)} \\ CC_3^2 = \frac{Cov(r_1,r_3)Cov(r_2,r_3)}{Cov(r_3,r_3)Cov(r_1,r_2)} \end{cases} \tag{7}$$

To clarify the MTC calculated RMSE with traditional evaluation, thereafter we only use RMSE that refers to MTC results and RMSD for traditional evaluation. Since both RMSE and CC are derived from covariance between the three products, the MTC statistics reveal the relative error, which is treated as uncertainties. Therefore, the less uncertain product or the best performance is the one that has the lowest RMSE and highest CC. Likewise, the most uncertain is associated with the highest RMSE and lowest CC. In the following content, the unit of RMSE is in logarithmic mm/hour and CC is unitless.

### 2.3.3. Data Preparation

The rainfall data retrieved from each source have different spatial extends and resolutions. To make them comparable, IMERG and MRMS are downscaled with nearest neighbors and aggregated to the same resolution as NCEP, with 4 km in space and 1-hour in time. To fit in MTC, the rainfall data should be logarithmically transformed. Previous studies demonstrated that the treatment of zero values can be done either by simply removing zero values [34,38] or replacing with near-zero values [37,40]. It is found that the displace with near-zero values can retain more sample sizes, especially in event analysis [40]. Based upon previous suggestions [37,40], the NAN values are then removed and zeros are replaced with $10^{-3}$ in this study. To obtain more robust results and filter out noise, the bootstrapping strategy is utilized with 500 trials for evaluation at each pixel and the mean values of RMSE and CC are used.

### 2.4. Conventional Statistical Metrics

A list of evaluation metrics is summarized in Table 2. As there is assumed no absolute ground truth in the extreme events, the "error" is termed as "difference" between two products, for example, Root Mean Squared Difference (RMSD). They are all computed in a domain at each pixel (4 by 4 km) for each pair. For categorical indices between two products, the Probability of Detection (POD), False Alarm Rate (FAR), Critical Success Index (CSI) are chosen as indicators to investigate the sensitivity of rainfall detection. The reasons that NCEP is not regarded as reference are: (1) NCEP is not well quality controlled; (2) it involves uncertainties from interpolation in space; (3) NCEP likely encounters device problems in extreme rainfall events. It thus exposes more uncertainties and is justified in our study analysis. The "reference" data in the denominator is chosen by the less uncertain product calculated with regard to MTC method. The threshold for computing these categorical indices is set to 0.1 mm/h [44].

Since it is also insightful to compute these metrics in a conditional way during extreme events [52], they are conditioned to regions within the core of Harvey to validate the consistency between unconditioned results and conditioned (i.e., RMSD, $RMSD_{cond,p}$). This condition is based upon what the percentile of rainfall rate the product exceeds with a similar formula in Table 2. To take the extreme values into account, three percentiles are selected (i.e., 50 percentiles, 75 percentiles, and 95 percentiles) with an analogy to previous work [52]. Therefore, at point-level, only rain rates beyond those conditions are used to compute the differences in the Harvey core.

**Table 2.** Continuous and categorical indices used in this study.

| | Metrics | Equation | Best Value | Conditional Values |
|---|---|---|---|---|
| Continuous Indices | Correlation coefficient | $\frac{\sum_{i=1}^{n}(x_i-\bar{x})(x_i-\bar{y})}{\sqrt{\sum_{i=1}^{n}(x_i-\bar{x})\sum_{i=1}^{n}(y-\bar{y})}}$ | 1 | $CC_{cond,p}$ |
| | RMS difference (RMSD) | $\sqrt{\frac{1}{n}\sum_{i-1}^{n}(x_i-R_i)^2}$ | 0 | $RMSD_{cond,p}$ |
| Categorical Indices | POD | $\frac{hits}{hits+misses}$ | 1 | $POD_{cond,p}$ |
| | FAR | $\frac{falsealarm}{falsealarm+hits}$ | 0 | $FAR_{cond,p}$ |
| | CSI | $\frac{hits}{falsealarms+hits+missies}$ | 1 | $CSI_{cond,p}$ |

Note: For categorical indices, hits is defined by both evaluated value and reference value larger than the threshold. The false alarm is an evaluated value larger than the threshold but a reference less than it. Misses are evaluated values less than the threshold while the reference is larger than it. Conditional metrics are defined by given source data larger than a threshold, i.e., p percentile of the daily rainfall for extreme events.

### 2.5. Hierarchical Evaluation

Three hierarchical levels of evaluation are performed: In level one, all events are combined (denoted as **All**) and further events breakdown as **Harvey** event only and **Other** non-Harvey events for cross-examination. In level two, an extreme Harvey event is exclusively taken into account to test

the consistency of MTC results with conventional metrics. Lastly, an analysis is conducted specifically within the storm core of Hurricane Harvey to examine uncertainty in such an ultra extreme region, with the core delineated with 400mm rainfall isoline. In the core region, the conditional metrics described above are also calculated, and several representative pixels are selected to investigate their temporal variations. These pixels from one to four, as shown in Figure 1, are: (1) Houston metropolitan area where NCEP is highly uncertain; (2) the maximum amount of rainfall observed by IMERG; (3) the maximum amount of rainfall observed by MRMS; and (4) the highly uncertain place for MRMS.

## 3. Results

### 3.1. Cross-Events Comparison

#### 3.1.1. Conventional Inter-Comparison

Figure 2 depicts the accumulative rainfall maps, and Table 3 lists the inter-comparison results with conventional metrics for three cases (i.e., All, Harvey, and Other) from the three datasets (i.e., NCEP, MRMS, and IMERG). The rainfall spatial patterns observed by each product are similar, as their spatial correlations are all above 0.8 cross cases in Table 3. MRMS/IMERG has the highest spatial correlation for All (0.91) and Harvey event (0.94) while similar to NCEP/MRMS in Other non-Harvey events (both 0.87). Despite these similarities, neither NCEP nor IMERG adequately capture the high spatial variability inside the core regions, due to their relatively coarse spatial resolutions. Especially in the Harvey case, one can observe the "patchiness" pattern for NCEP, which is further investigated in Section 3.3. From the rainfall distribution histogram inserted in Figure 2, the non-Harvey events have higher densities of rainfall amount in the light rain range (<200 mm), but Harvey produces more in the heavy rain range (>800 mm).

Table 3 lists the inter-comparison results with conventional metrics. In terms of total accumulation, MRMS always observes the highest accumulative rainfall amount, followed by IMERG and NCEP. Both spatial and temporal correlations demonstrate the highest similarity between the two remote sensing technology-derived IMERG and MRMS in all cases. For categorical metrics (i.e., POD, FAR, CSI), IMERG and MRMS also show the best agreement in general, suggesting NCEP possibly suffers from gauge device malfunction in these extreme events.

**Table 3.** Overall conventional statistics for three cases (i.e., All, Harvey, and Other) and pairs of comparison.

| Metrics | | All | Harvey | Other |
|---|---|---|---|---|
| | NCEP | 1366 | 979 | 686 |
| Max. total rain (mm) | MRMS | 2876 | 1625 | 1451 |
| | IMERG | 1749 | 1116 | 853 |
| | NCEP/IMERG | **2.51** | 1.44 | **1.87** |
| RMSD (mm/h) | NCEP/MRMS | 3.08 | 1.55 | 2.54 |
| | IMERG/MRMS | 2.71 | **1.42** | 2.20 |
| | NCEP/IMERG | 0.52 | 0.48 | 0.48 |
| Temporal CC | NCEP/MRMS | 0.49 | 0.54 | 0.40 |
| | IMERG/MRMS | **0.62** | **0.56** | **0.60** |
| | NCEP/IMERG | 0.85 | 0.90 | 0.85 |
| Spatial CC | NCEP/MRMS | 0.80 | 0.87 | **0.87** |
| | IMERG/MRMS | **0.91** | **0.94** | **0.87** |
| | NCEP/IMERG | 0.64 | 0.52 | 0.56 |
| POD | NCEP/MRMS | **0.72** | 0.58 | 0.62 |
| | IMERG/MRMS | 0.70 | **0.69** | **0.62** |
| | NCEP/IMERG | 0.28 | 0.47 | 0.39 |
| FAR | NCEP/MRMS | 0.29 | 0.39 | 0.42 |
| | IMERG/MRMS | **0.19** | **0.41** | **0.27** |

**Table 3.** *Cont.*

| | Metrics | All | Harvey | Other |
|---|---|---|---|---|
| | NCEP/IMERG | 0.52 | 0.44 | 0.41 |
| CSI | NCEP/MRMS | 0.57 | **0.52** | 0.43 |
| | IMERG/MRMS | **0.60** | 0.51 | **0.51** |

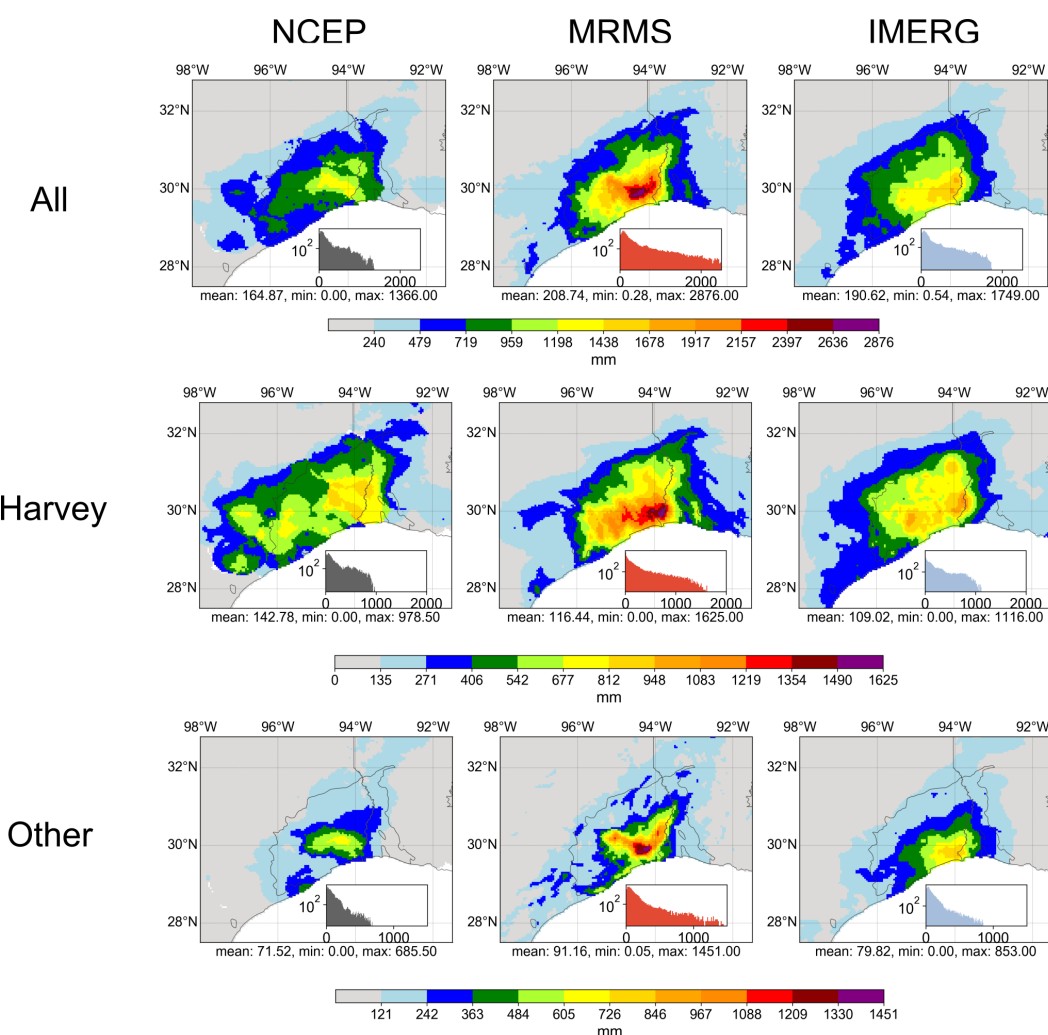

**Figure 2.** Accumulative rainfall observed in three cases. Inside each axes, the inset corresponds to the histogram of accumulative rainfall.

### 3.1.2. MTC Comparison

Because no absolute ground truth is assumed, MTC instead evaluates the relative deviation from "truth" among three independent products. The first overview of TC analyzed results for All, Harvey, and Other are illustrated in Figure 3. Both CC (relative agreement) and RMSE (relative deviation) indicate MRMS (CC = 0.80; RMSE = 2.77 mm/h) is the overall best product (closest to truth) across all events, followed by IMERG (CC = 0.72; RMSE = 4.35 mm/h) and then NCEP (CC = 0.66; RMSE = 5.16 mm/h). Therefore, the similarity and difference of NCEP and IMERG relative to the MRMS are mostly notable in the storm core Houston area, especially during the Harvey extreme event. For example, NCEP differs from MRMS more in the Houston metropolitan areas, as marked in black circles in Figure 3b. This result is on a par with the investigation of Chen et al. (2020) [53] in the Harvey event that MRMS has the highest CC (0.91) value and correspondingly lowest RMSE (5.75 mm/h) among NCEP gauge-only products and IMERG V06A in Hurricane Harvey. IMERG, on the other hand,

exhibits better agreements with MRMS in the rainfall core regions than NCEP. Notably, the deviation from the most uncertain NCEP to the best overall MRMS is much more magnificent in the Harvey case, which can also be quantified as the range of the median value of CC (0.14, **0.27**, and 0.09) and RMSE (1.79, **2.01**, 1.52 mm/h) for All, **Harvey**, and Other, respectively. This reveals that MTC can capture the high variability in this 500-year return Harvey extreme event. In summary, the above analysis results from both the MTC method and the conventional metrics suggest the consistence and robustness of TC in extreme events.

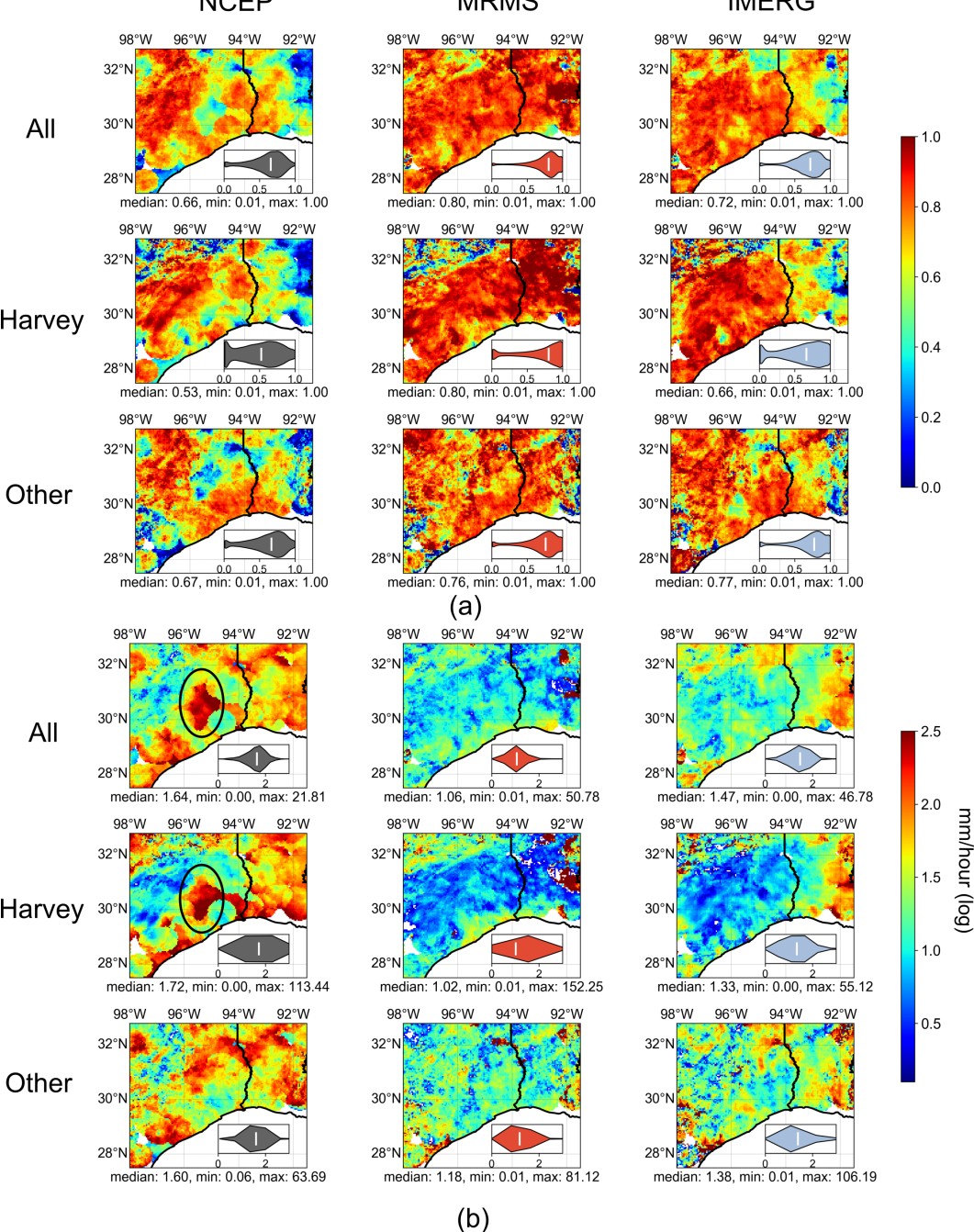

**Figure 3.** Spatial plot of Multiplicative Triple Collection (MTC) results (**a**) Correlation Coefficient (CC); (**b**) Root Mean Squared Error (RMSE) for three cases (concatenated all events, Harvey-only and non-Harvey) in each row, and each column represents each product. The small panel inside each axis is the violin plot for the metric, and the white bar is where the median value is located. Two marked circles emphasize where the National Centers for Environmental Prediction (NCEP) is highly uncertain.

### 3.2. Hurricane Harvey Analysis

#### 3.2.1. Conventional Inter-Comparison and MTC Results

The excessive rainfall extreme and high diverging behaviors of the TC results of Hurricane Harvey in the previous results motivate us to further investigate this event at the pixel level, as shown in Figure 4. First, the conventional CC indicates the MRMS/IMERG pair has the highest correlation values (0.69) with the least range than other pairs; and MTC calculated CC is also ranked MRMS (0.86), IMERG (0.79), and NCEP (0.65) from high to low. Second, even though conventional RMSD for each pair is similar (2.23 mm/h for NCEP/MRMS and NCEP/IMERG, 2.16 mm/h for MRMS/IMERG), MRMS/IMRG (NECP/MRMS) has the smallest (largest) range; similarly, MTC ranks MRMS to have the least (2.25 mm/h) , followed by IMERG (2.60 mm/h) and NCEP (4.80 mm/h) based on RMSE. A previous study by Omranian et al. (2018) [54] revealed the average POD, FAR, and CSI score to be 0.9, 0.3, and 0.7, respectively, for the IMERG Final product against radar data in the Harvey case. This result is consistent with our study (POD = 0.92; FAR = 0.25; CSI = 0.73). Overall, IMERG/MRMS pair has the highest agreement among categorical indices. Again, the results at point-scale corroborate the consistence and robustness of MTC in this extreme Harvey event.

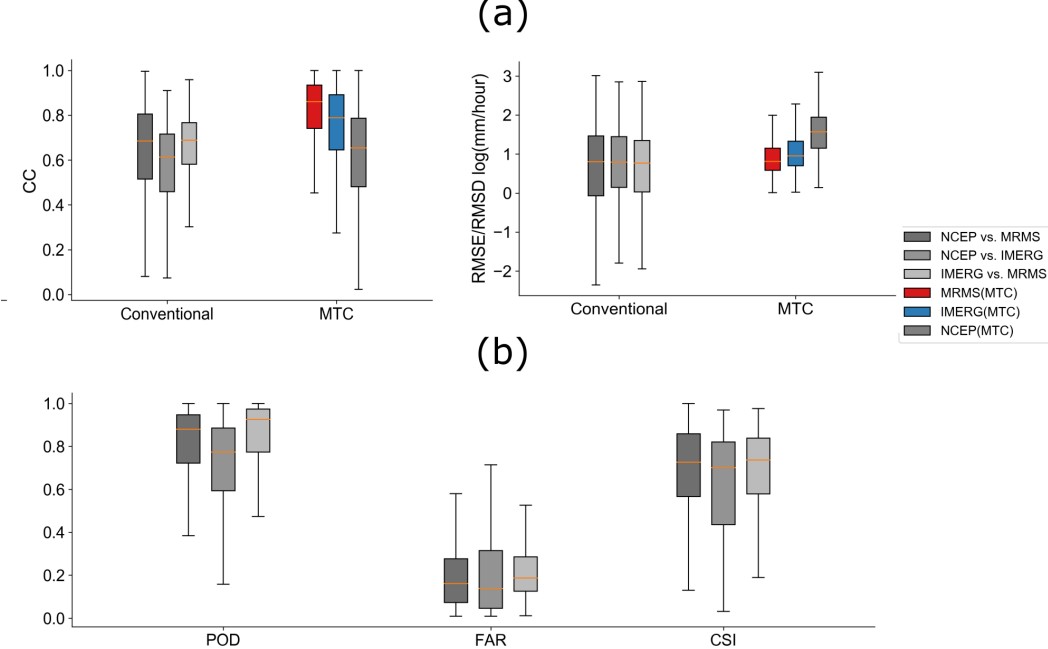

**Figure 4.** Boxplot for conventional metrics and MTC results in Hurricane Harvey. (**a**) Continuous indices and MTC results; (**b**) categorical indices.

#### 3.2.2. Further Exploration of MTC Results

Since both cross-events results and exclusive Harvey case show the applicability of MTC in extreme events in previous sections, MTC is further applied to investigate the performance of each product with respect to accumulative rainfall ranges at 50 mm intervals, as shown in Figure 5. Overall, MRMS performs consistently well, with higher CC cross the range and lower RMSEs (only slightly higher than IMERG at a higher range 1100 vmm). Interestingly, IMERG shows higher uncertainties in the light rain (i.e., below 150 mm), but with improved performance gradually toward a higher range. This is similar to reported literature [55–57]. Specifically, Omranian  et al. [12] also concluded that the IMERG final product generally has better performance with higher precipitation rates compared to lower rates in the case of hurricane Harvey.

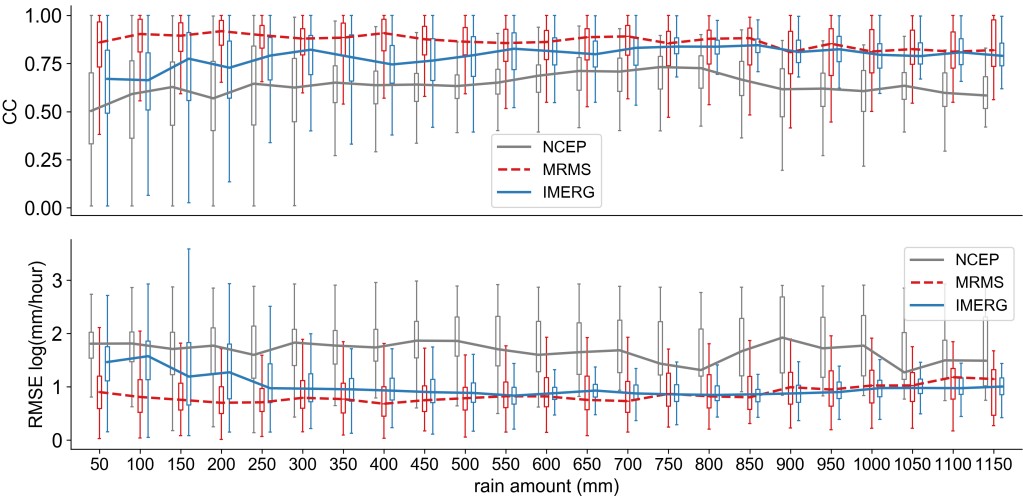

**Figure 5.** Boxplot of MTC measured CC, RMSE (from top down) in Hurricane Harvey at each accumulative rain bins for a 50 mm interval based on MRMS data. The lines connected the median value of each box for the corresponding product.

### 3.3. Storm Core of Harvey Event

Figure 6 depicts the MTC results of CCs and RMSEs for whole area, storm core, and non-core of Harvey with the core delineation of 400 mm rain isoline. First, the results (i.e., RMSEs and CCs) are generally better inside the core region because the rainy samples inside the core are more than non-cores and thus strengthen the robustness of MTC. Second, the performance for each product remains consistent: MRMS > IMERG > NCEP. Therefore, recall previous results, MTC gives consistent rankings of these products in three different levels: Cross-events, Harvey event, and Harvey storm core.

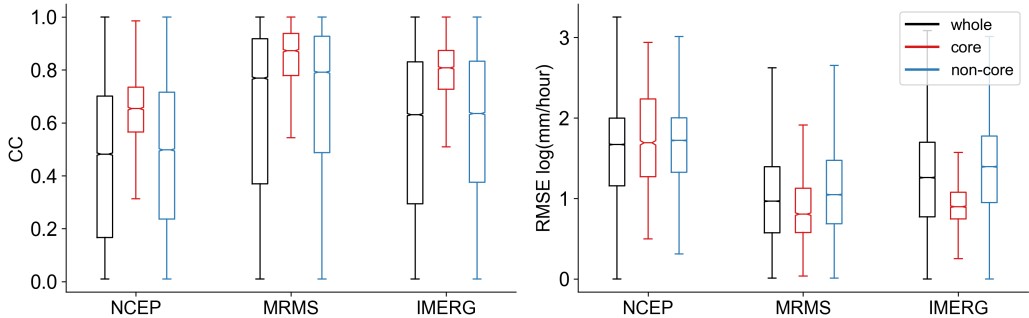

**Figure 6.** Boxplot of MTC measured CCs and RMSEs for three products grouped by whole, core, and non-core regions in Hurricane Harvey. Extreme values, i.e., outliers (outside of 1st/3rd quartile ∓ inter-quartile), are ignored to visualize the difference. The notch represents the median value for the samples in the region.

By looking at the core of Harvey alone, the accumulative rainfall distribution in Figure 7 reveals the systematic difference in this 500-year return extreme event. NCEP has the highest density within the low range of 400 to 600 mm, contributing almost 60% of the occurrence. In this same low range, IMERG with MRMS illustrates more similarities. In the middle range (i.e., 650 to 900 mm), on the contrary, NCEP and MRMS achieve better agreements and IMERG seems to overdetect. Unlike NCEP and IMERG, which drop off after 1100 mm, MRMS continuously observes rainfall in the higher end. Hence, both NCEP and IMERG underestimate high-end rainfall and overestimate at the lower/middle range.

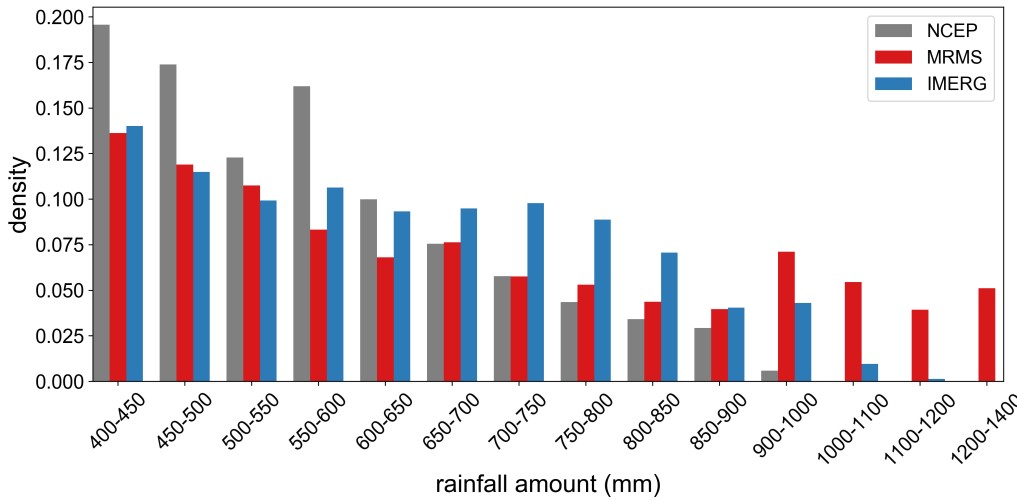

**Figure 7.** Histogram of accumulative rainfall binned at 50 mm interval from 400 (core) to 1400 mm in Hurricane Harvey. Vertical axis indicates the density.

Figure 8 further examines their performance conditioned at the medium to high end accumulative rainfall at three different percentiles (50th, 75th, and 95th). Figure 8a shows the conventional metrics with consistent results to MTC results across the three percentiles. Moreover, the differences of categorical indices in Figure 8b indicate IMERG/MRMS performs the best among the three pairs. In general, these results (conditional high rainfall) again approve the applicability of MTC in (ultra) extreme events.

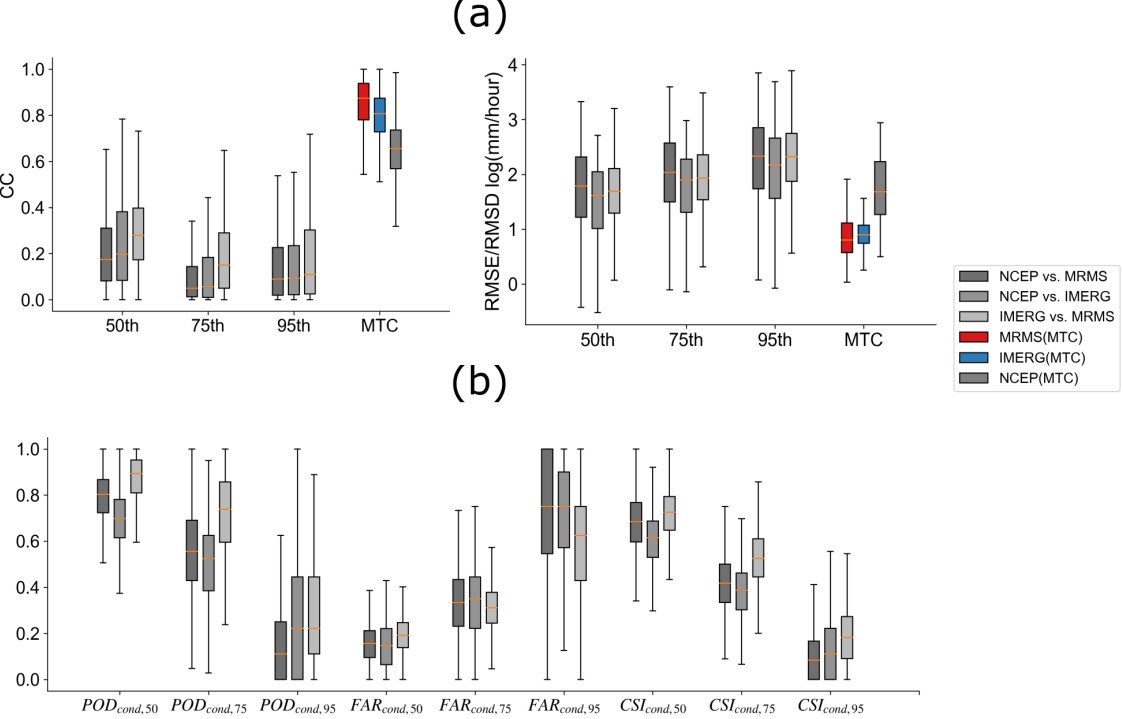

**Figure 8.** Boxplot of metrics conditioned at 50, 75, and 95 percentiles in the core region: (**a**) Continuous indices; (**b**) Categorical indices.

To look into event temporal variability, the time series of storm core average and some representative pixels of extreme are shown in Figure 9. Overall, the areal average time series in the uppermost panel shows they all indeed captured the temporal variability of the storm event.

However, there are variations in pixel (local) details for each product. Site 1, 3, and 4 refer to the place where NCEP have very high uncertainty (MTC calculated RMSE beyond 8 mm/h). The gray windows in the three corresponding time series plots indicate when NCEP data either showed zero value (site 1, 3) or stopped recording any data (site 4), while the other two remote sensing data sources captured intense rainfall. Site 2 is selected as IMERG measured the maximum amount of rainfall in this event. The horizontal blue line marks the upper maximum rain rate of 60 mm/h that GMI can observe due to its sensor sensitivity [49]. In other words, the IMERG product has an upper bound due to its sensor sensitivity. The red-shaded windows in site 1, 3, and 4 indicate when MRMS rises up instantaneously while NCEP and IMERG show some degree of agreements at lower rain rates. Such spike anomalies could be caused either by non-weather echoes, or more likely high-sensitivities in the Z-R relation to hails mixed with rain storms [21,58].

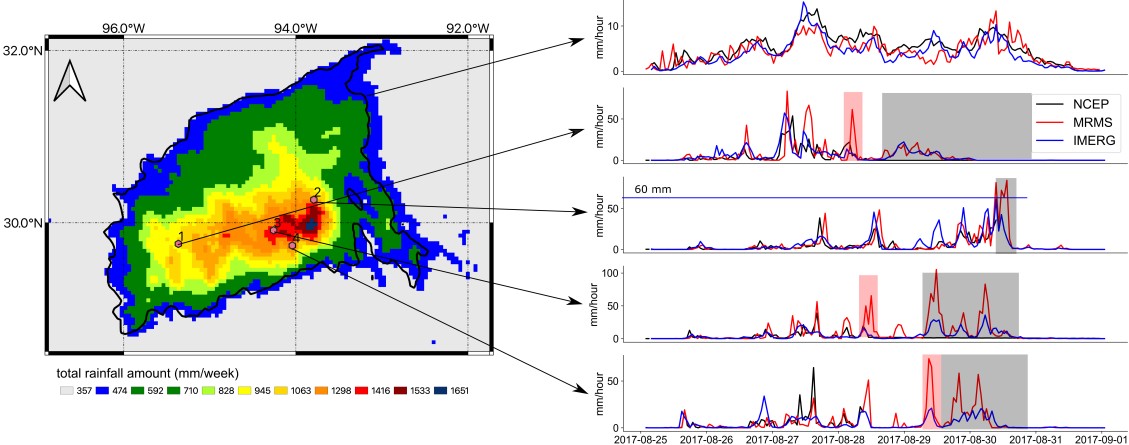

**Figure 9.** Core separated plot with selected pixel-wise time series. The black thick line in the left panel is the 400 mm accumulated rainfall contour line to separate the Harvey core regions. The windows in the right pane highlight special characteristics, with black corresponding to the NCEP observation, red for MRMS, and blue for Multi-satellitE Retrievals for GPM (IMERG).

## 4. Discussions

Throughout this study, three state-of-the-art precipitation products are intercompared with the conventional and MTC method. It is worth noting that during extreme events, particularly for 500-year return Harvey event, the ground-based observations (i.e., gauge and radar) are possibly contaminated due to saturation/malfunction and hot signals. Observations from space (i.e., satellite) suffer from relatively coarse spatiotemporal resolution and is thus unrepresentative of the storm core. Previous studies [32,54] described in detail the impact of spatiotemporal resolution to the accuracy of rainfall products. With the limitation on identifying "ground truth", this study adopts the MTC method and complements with the conventional method to provide the evaluations of the three products in extreme events. These agreements between MTC and conventional metrics corroborate these research findings and suggest the potentials of the MTC method in such circumstances. Beyond that, it is opportunistic to utilize the results of MTC and further develop innovative data merging/assimilation methods to integrate for one best possible combined product, especially suitable for extreme events.

## 5. Conclusions

In order to better understand uncertainties of precipitation estimation in extreme events, this study utilizes both traditional metrics and also the MTC method to cross-examine the similarities and differences of three independent rainfall datasets, i.e., MRMS, IMERG, and NCEP in Houston Storm/Hurricane events during the GPM era.

First, the applicability of the MTC method to extreme events is cross-evaluated with the conventional metrics and the results reveal that: (a) the consistency of cross-examination results against traditional metrics (both conditioned and non-conditioned continuous/categorical metrics) approves the applicability of MTC in multiple extreme events; (b) the consistency of cross-event evaluation results also indicates the robustness of the MTC method across such individual extreme events.

Second, both traditional metrics and especially MTC results conclude that: (c) all three independent products demonstrate their capacity of capturing the spatial and temporal variability of the storm structures while also magnify respective inherent deficiencies; (d) NCEP and IMERG likely underestimate, while MRMS overestimates the storm total accumulation, especially for the 500-year return Hurricane Harvey; (e) both NCEP and IMERG underestimate extreme rain rates (>= 90 mm/h), while MRMS maintains robust performance across the rain storm range; (g) all three products show respective inherent deficiencies in capturing the storm core of Harvey. For example, the NCEP discontinuity in time (zero or no records) is likely due to in-situ tipping buck gauge saturation or device malfunctions during the Harvey extreme; the relative coarse spatiotemporal resolution of IMERG under-capture the extremes; and while MRMS experiences "hot" radar signals possibly from non-weather echoes or hail contamination during such extremes.

Given the unknown ground reference assumption of MTC, this study suggests that MRMS has the best overall performance, followed by IMERG and NCEP. The similarities, differences, advantages, and deficiencies revealed in this study could guide the users, such as operational agencies, to carefully select products for emergency planning and response during extreme storms, and also hopefully motivate the research community to improve respective gauge network designs and sensor/algorithms. Furthermore, this study also calls for developing innovative multi-platform multi-sensor data merging/assimilation methods to integrate one best possible combined product, especially suitable for extreme storm events.

**Author Contributions:** Conceptualization, Z.L., S.G., and Y.H.; data curation, Z.L., M.C. and Z.H.; methodology, Z.L. and Y.H.; software, Z.L.; validation, Z.L., M.C., G.T. and Z.H.; formal analysis, Z.L.; investigation, Z.L.; resources, Z.L., Y.H. and J.J.G.; writing—original draft preparation, Z.L.; writing—review and editing, S.G., Y.H., J.J.G., G.T. and Y.W.; visualization, Z.L.; supervision, Y.W., J.J.G. and Y.H.; project administration, Y.H.; funding acquisition, Y.H. All authors have read and agreed to the published version of the manuscript.

**Funding:** This research received no external funding.

**Acknowledgments:** We would like to acknowledge the efforts made by NOAA/NSSL and the NASA science team for making MRMS and IMERG precipitation data accessible. The first author is supported by the University of Oklahoma Hydrology and Water Security (HWS) program (https://www.ouhydrologyonline.com/) and Guoqiang Tang is funded by Global Water Future.

**Conflicts of Interest:** The authors declare no conflict of interest.

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
