# Peer review of "Cross-Examination of Similarity, Difference and Deficiency of Gauge, Radar and Satellite Precipitation Measuring Uncertainties for Extreme Events Using Conventional Metrics and Multiplicative Triple Collocation"

_remotesensing, doi:10.3390/rs12081258_

Round 1
Reviewer 1 Report
The paper is mainly about precipitation measurements based on different sources and their comparison. This study investigates these products’ performance during extreme events such as hurricanes, which is a hot topic and really needed to be more discussed in the scientific communities.
It is very well-organized and well-written while it still needs some minor changes before final decision for publication as follows:
Briefly discuss how spatial resolution of your data can change the accuracy of your analysis. Recent studies show that changes in spatio-temporal resolution of satellite products can highly affect the accuracy of the analysis. A study by Omranian and Sharif entitled “Evaluation of the Global Precipitation Measurement (GPM) satellite rainfall products over Lower Colorado River basin, Texas”, indicates this point very comprehensively. Try to put more detail about this very important fact.
You have generated very good results, and mainly compare your results together. However, my suggestion is that it would be nice if you also compare yours with other studies output since you mentioned that there are other studies which used the same method. As an example, POD, FAR and CSI are three indices that are described a lot in other similar studies. Please add a few comparisons to your text.
Author Response
- Briefly discuss how spatial resolution of your data can change the accuracy of your analysis. Recent studies show that changes in spatio-temporal resolution of satellite products can highly affect the accuracy of the analysis. A study by Omranian and Sharif entitled “Evaluation of the Global Precipitation Measurement (GPM) satellite rainfall products over Lower Colorado River basin, Texas”, indicates this point very comprehensively. Try to put more detail about this very important fact.
Yes, spatiotemporal resolution plays an essential role and the conclusion “the relative coarse spatiotemporal resolution of IMERG under-capture the extremes” indicates this point. To emphasize this, we added the reference you suggested and text in the discussion session:
“Observations from space (i.e., satellite) suffers from relatively coarse spatiotemporal resolution and is thus unrepresentative of the storm core. Previous studies (Hong et al., 2006; Omranian and Sharif, 2018) in detail described the impact of spatiotemporal resolution to the accuracy of rainfall products.”
- You have generated very good results, and mainly compare your results together. However, my suggestion is that it would be nice if you also compare yours with other studies output since you mentioned that there are other studies which used the same method. As an example, POD, FAR and CSI are three indices that are described a lot in other similar studies. Please add a few comparisons to your text.
Added comparison of my results with other studies in line 273-275:
“This result is on a par with the investigation of Chen et al. (2020) in Harvey event that MRMS has the highest CC (0.91) value and correspondingly lowest RMSE (5.75 mm/h) among NCEP gauge-only products and IMERG V06A in Hurricane Harvey”
In line 290:
“Previous study by Omranian et al. (2018) revealed the average POD, FAR, and CSI score to be 0.9, 0.3, and 0.7, respectively for IMERG Final product against radar data in Harvey case. This result is consistent with our study (POD=0.92;FAR=0.25;CSI=0.73).”
Reviewer 2 Report
Review Comments for "Cross-examination of Similarity, Difference and Deficiency of Gauge, Radar and Satellite Precipitation Measuring Uncertainties for Extreme Events using Conventional Metrics and Multiplicative Triple Collocation" Li et al.
This paper focuses on quantifying uncertainties of precipitation estimations for extreme cases from three different platforms, including rain gauges, radars, and satellites. Both traditional metrics and new method MTC are used in the paper.It shows the potential of using MTC method for independent datasets. The paper is well written and easy to follow. The figures are well done. I only have a few minor suggestions.
- Line 63: "error in Z-R relations", I think it is not a 'error', it is more like the uncertainty of applying Z-R relationship to different types of precipitation in different climate regions.
- Figure 1: What are those numbers on the right bottom (1-4) mean on the plot?
- Table 1: Are the maximum R for Bill and Imelda are missing? why?
- Line 260: why MRMS always observes higher R overall?
- Figure 5: I like this plot, but I would suggesting replotting this for rain amount bin in log scale, like (1, 10, 100, 1000 mm). I think it will show more clearly the trend.
